# Simulation Study on the Characteristics of Gas Extraction from Coal Seams Based on the Superposition Effect and Hole Placement Method

Jin Yan [ID], Minbo Zhang [ID], Weizhong Zhang * and Qinrong Kang

School of Resource and Security Engineering, Wuhan Institute of Technology, Wuhan 430074, China; yanjin19982001@163.com (J.Y.); zhangminbo2015@163.com (M.Z.); kangqinrong02@163.com (Q.K.)
* Correspondence: zhangweizhong2023@163.com

**Abstract:** In order to obtain a reasonable extraction drilling method for coal seam working faces and to carry out targeted as well as cost-effective hole placement optimization, a gas–solid coupled model based on the coal rock deformation field and the matrix–fissure dual seepage–diffusion field was established and numerically solved via the use of COMSOL Multiphysics finite element software to optimize the gas transport parameters of the Dongpang coal mine based on the study of the coal seam gas transport law. This study shows the following: With an increase in the extraction time, the gas content of the coal seam was reduced to a minimum. It shows that, with an increase in the extraction time, the gas pressure and seepage velocity keep decreasing the stable value, the main stress around the borehole redistributes, and the coal permeability keeps decreasing with an increase in the decay coefficient. The extraction radius of the boreholes increases exponentially with the extraction time, and the reasonable extraction hole size is 94 mm; the use of multiple boreholes for pre-drawing gas via the use of the interval between the effective extraction radius and the influence radius results in a blind extraction zone caused by the superposition effect between the boreholes, and the residual gas value of the coal seam increases with an increase in the borehole spacing. The use of an equilateral triangular hole layout can avoid the extraction blind zone, and the pre-sumping effect is good after field application, which ensures the safe production of coal mines as well as the reasonable use of resources.

**Keywords:** gas extraction; superposition effect; numerical simulation; effective extraction radius; gas transport; hole placement

## 1. Introduction

Due to the specificity of China's energy structure, the demand for coal mines is increasing, and mining is accompanied by the risk of coal and gas protrusion [1]. Gas extraction is an effective measure with which to prevent such mine accident disasters and is the main means with which to manage and utilize gas in China [2,3]. At present, through a combination of field practice and numerical simulation, it is the main technical means with which to improve extraction efficiency and ensure stable gas extraction by mastering the law of coal seam gas transport and determining the effective extraction radius of boreholes [4,5], so as to provide a reference basis for a reasonable layout of drilling methods and spacing.

Coal is a very complex porous medium, of which the permeability of coal is considered to be a key factor in characterizing the reservoir capacity of a coal seam [6,7]. Therefore, establishing a reasonable gas permeability model and revealing the gas transport law can provide a reasonable theoretical basis for gas extraction volume prediction and extraction engineering design. Numerous scholars have obtained rich research results for the diffusion permeability model, flow mechanism, and mechanical properties of gas-bearing coal. Among the studies on gas–solid coupling models of coal and gas, Liang et al. [8] constructed a flow–solid coupling model for gas extraction based on the intrinsic equations of the coal

body and the equations of the seepage field, taking into account the influence of the deformation of the coal rock body and the adsorption as well as desorption processes of gas in the pores and fissures on the seepage characteristics. Li et al. [9] collected coal samples with different degrees of metamorphism from different mining areas for coal grain gas diffusion experiments and found that the experimental data were larger than the theoretical data via comparing the experimental results with the theoretical analysis values, in addition to establishing a multiscale time-varying diffusion model based on multiple pore structures that laid the foundation for the study of permeability models. Yue et al. [10] established a Klinkenberg effect consideration based on the extraction characteristics of low-permeability coal seams based on the influence of the Klinkenberg effect, the coal skeleton, and the matrix shrinkage effect on gas extraction. Liu et al. [11,12] considered gas transport as a "double-porosity–single-permeability" system and modified the Palner-Mansoori model. Wang et al. [13] developed a coupled model that considered the diffusion of adsorbed methane from a matrix to pore space based on the dual-porosity–single permeability model, and described the relationship between particle deformation, gas flow, and gas pressure in coal seams. Tan et al. [14] studied the coal double-dispersion model in depth on the basis of single-pore diffusion, and their study achieved better results compared to single-pore diffusion. Zhao [15] developed a gas seepage model based on the gas mass conservation equation, Darcy's law, Langmuir's analytical equation for sorption, Klinkenberg's equation, the effective stress equation, and the pore permeability kinetics equation to investigate the effect of gas seepage patterns between extraction boreholes on gas seepage. Chen et al. [16] introduced surface stresses into the equation for effective stresses in coal seams based on the basis of dual-porosity elastodynamics, and proposed a mechanism for the response of fracture–matrix interaction to porosity, taking into account the effects of seepage and deformation.

With the above models proposed, mastering the gas transport law through numerical simulation software is the mainstream approach, especially by verifying the characteristics of the gas extraction radius to provide an optimized solution for on-site extraction. Qi et al. [17] established an expression for the pressure in a coal seam around a gas extraction borehole, which provided a basis for subsequent simulations to calculate effective radii; Hao et al. [18] considered the rheological properties of coal and established a coupled model of adsorption, seepage, and stress in a coal seam to determine effective gas extraction radii through the law of change in permeability during gas extraction. Guo et al. [19] established a gas–solid coupling model for the coal seepage of coal seam gas, verified the relationship between the borehole diameter, the extraction time, the extraction negative pressure, and the effective extraction radius, and proposed the concept of a pressure difference enhancement ratio. Li et al. [20,21] considered the seepage mechanism of coal matrix gas and studied the coal seam gas transport law by establishing a coupled field that considered coal matrix gas seepage and coal rock deformation, and concluded that the matrix gas seepage was less than the diffusion effect. Cheng et al. [22] studied the law of gas transport in the overburden fracture zone under the influence of mining, and designed reasonable parameters for high-level directional long holes for high-pressure gas extraction in an extraction zone based on the test results. Xu et al. [23] considered coal to be a double-porous elastic medium and studied the gas transport law based on the dual permeability of a matrix and a fracture, and achieved good simulation results in the field. Zou et al. [24] studied the effective extraction radius of gas with the use of COMSOL in order to improve the efficiency of gas extraction and determined the spacing of hole placement in practical engineering. Zhang et al. [3] established a fully coupled model that considered coal damage deformation via dividing the permeability change stages by plastic damage extent, and studied the influence of different factors on the effective extraction radius to optimize the gas discharge design of a specific coal mine working face. Xu et al. [25] proposed reasonable hole spacing and spacing based on the study of effective extraction radii to reduce the influence of the superimposed extraction effect of boreholes in the process of pre-pumping gas from coal seams, and improved the efficiency of gas extraction by establishing a coal seam gas–solid

coupling model. Wei et al. [26] studied a cross-test of a drilling coefficient and coal seam geological factors through the use of COMSOL to study the effect of combined multi-drill hole extraction on attainment areas, and proposed the method of dividing the extraction target area according to different geological conditions in order to make the gas extraction more accurate. In order to explore the gas extraction characteristics of double boreholes in a cis-seam, Guo et al. [27] established a gas–solid coupling model for gas-bearing coal based on the consideration of adsorption, expansion, and the Klingberg effect, in addition to studying the reasonable spacing of boreholes for specific coal seams.

In summary, numerous scholars have proposed mathematical models based on gas extraction and achieved corresponding results in field practice through the method of numerical simulation. As most studies base their models on the condition that a coal body is double-porous and single-permeable, in view of this, the authors construct equations for the diffusion–percolation fields of gas in a coal matrix and fractures as well as the deformation fields of a coal body based on the assumption that a coal body is an elastic medium, take full account of the characteristics of a coal body being double-porous and double-permeable, establish a gas–solid coupling model for gas-bearing coal seams, and solve said equations through using COMSOL. After determining the effective extraction radius and gas transport law of single-hole extraction, the reasonable spacing and placement of multi-hole extraction are then investigated in order to avoid blind areas caused by superimposed extraction, with a view to providing some guidance on coal mine gas extraction work.

## 2. Flow–Solid Coupling Model for Gas Extraction
### 2.1. Equation for Dynamic Changes in Permeability

Under the influence of fracture gas pressure and matrix pore pressure, the matrix porosity and fracture porosity of dual-pore structure coals can be shown as follows [28]:

$$\frac{\varphi_m}{\varphi_{m_0}} = \frac{1}{(1+S)}\left[(1+S_0) + \frac{\alpha_m(S-S_0)}{\varphi_{m0}}\right] \tag{1}$$

where $\varphi_m$ is the matrix porosity and $\varphi_{m_0}$ is the initial matrix porosity. The coal matrix porosity strain is $S = \varepsilon_V + \frac{P_m}{K_s} - \varepsilon_L\frac{P_m}{P_L+P_m}$. The initial matrix pore strain is $S_0 = \varepsilon_{V0} + \frac{P_{m0}}{K_s} - \varepsilon_L\frac{P_{m0}}{P_L+P_{m0}}$. $\alpha_m$ is the substrate Biot factor.

$$\frac{\varphi_f}{\varphi_{f0}} = 1 - \frac{3}{\varphi_{f0} + \frac{3K_f}{K}}\left(\frac{\varepsilon_L\Delta p_m}{p_L + \Delta p_m} - \varepsilon_v\right) \tag{2}$$

where $\varphi_f$ is the fracture porosity and $\varphi_{f0}$ is the initial fracture porosity; $K_f$ is the fracture stiffness, MPa; $K$ is the bulk modulus of the coal body, MPa; $\Delta p_m$ is the matrix pressure variation value, MPa; $P_L$ is the Langmuir pressure, MPa; and $\varepsilon_v$ is the bulk strain of the coal.

Considering only the elastic deformation of the coal body, the relationship between coal permeability and porosity is in accordance with the cubic law [29]:

$$\frac{k}{k_0} = \left(\frac{\varphi}{\varphi_0}\right)^3 \tag{3}$$

where $k$ and $k_0$ represent the coal seam permeability and initial permeability, respectively, mD; $\varphi$ and $\varphi_0$ represent the coal seam porosity and initial porosity, respectively.

The coal matrix permeability, $k_m$, can be expressed through the following equation [30]:

$$k_m = k_{m0}\left(\frac{1}{(1+S)}\left((1+S_0) + \frac{\alpha_m(S-S_0)}{\varphi_{m0}}\right)\right)^3 \tag{4}$$

where $k_{m0}$ is the initial permeability of the matrix, mD. $k_{m0}$ is the initial permeability of the matrix, mD.

The influence of gas flow pressure should be considered in fracture systems. The Klinkenberg effect exists in the flow of gas in porous media, and the smaller the gas pressure the more obvious the effect is; when the pressure tends toward infinity, the effect ceases, at which point the corresponding fracture permeability is the absolute permeability [31,32]. Therefore, the relationship between the effective permeability of gas and the absolute permeability should not be neglected. Equation (5) shows the dynamic evolution of fracture permeability with the introduction of Klinkenberg:

$$k_f = k_{f0} \left( 1 - \frac{3}{\varphi_{f0} + \frac{3K_f}{K}} \left( \frac{\varepsilon_L \Delta P_m}{P_L + \Delta P_m} - \varepsilon_V \right) \right)^3 \left( 1 + \frac{K_b}{P_f} \right) \tag{5}$$

where $k_f$ is the coal seam permeability, mD; $k_{f0}$ is the initial permeability of the fissure, mD; $K_b$ is the Klinkenberg coefficient, generally taken as 0.251; and $p_f$ is the fissure gas pressure, MPa.

### 2.2. Matrix Pore Gas Diffusion Equation

Both adsorbed and free gas are present in the pore system of the coal matrix [33], and the gas content of the matrix system, mm, is as follows:

$$m_m = \frac{V_L p_m \rho_c M_C}{(p_L + p_m) V_M} + \varphi_m \frac{M_C p_m}{RT} \tag{6}$$

where $\rho_c$ is the pseudo-density of the coal body, kg/m$^3$; $M_c$ is the molar mass of methane, kg/mol; $R$ is the gas constant, J/(mol·K); $T$ is the temperature of the coal seam, K; and $V_M$ is the molar volume of the gas at standard conditions, L/mol.

The equilibrium state of the coal bed matrix system and the fissure system is broken under the action of the pressure difference, and the diffusion of adsorbed gas in the matrix provides a mass source for the fissure gas transport due to the different velocities of matrix diffusion and fissure seepage [33], which can be obtained from the mass conservation theorem:

$$\frac{\partial m_m}{\partial t} = -\zeta \frac{D_t M}{RT} \left( p_m - p_f \right) \tag{7}$$

where $D_t$ is the coal seam diffusion coefficient, m$^2$/s, $D_t = D_0 \exp(-\lambda t)$; $\lambda$ is the attenuation factor, s$^{-1}$. $\zeta$ is a matrix shape factor, 1/m$^2$.

The dispersion equation for the matrix gas is obtained by combining Equations (6) and (7):

$$\frac{\partial p_m}{\partial t} = -\frac{3\pi^2 V_m (1 + b p_m)^2 (p_m - p_f) D_t}{L^2 \left[ a b \rho_c RT + \varphi V_m (1 + b p_m)^2 \right]} \tag{8}$$

where $a$ is the Langmuir volume product, m$^3$/t; $b$ is the adsorption equilibrium constant, MPa$^{-1}$; and $L$ is the coal body fissure spacing, m.

### 2.3. Fractured Gas Seepage Characteristics

The form and size of coal seam fractures are heterogeneous, and studies have concluded that the flow of gas in fracture systems is linear seepage, following Darcy's law [34,35]:

$$v = \frac{k}{\mu} \nabla P_f \tag{9}$$

where $v$ is the rate of gas seepage, m/s; $k$ denotes the coal seam permeability, Md; $\mu$ denotes the dynamic viscosity coefficient of gas, generally taken as $1.08 \times 10^{-5}$, Pa·s; and $\nabla P_f$ is the gradient of fracture pressure variation, MPa.

The amount of change in gas in the fissure of a coal body is equal to the difference between the gas flowing into the fissure and the gas flowing into the extraction borehole [31], as follows:

$$\frac{\partial}{\partial t}\left(\varphi_f \frac{M_g}{RT} P_f\right) = \left(1 - \varphi_f\right)\frac{D_t M_g}{RT}\left(P_m - P_f\right) - \nabla \cdot \left(-\frac{M_g}{RT}p_f v\right) \tag{10}$$

### 2.4. Control Equations for the Deformation of Coal Rock Masses

Assuming that a coal body is an isotropic elastic medium, there are differences in the effective stresses acting on the coal skeleton due to the dual structural nature of the coal seam, with different gas pressures between matrix pores and fissures [25]. By quoting the Biot coefficient into the Taisha base effective stress equation and introducing the effect of adsorption expansion stress on the effective stress in the coal skeleton [26], the controlling equation for the variable stress field in the coal seam, considering the coal-matrix-induced volume strain Navier form correction, can be obtained as follows:

$$\sigma_a = \frac{a\rho_s RT\ln(1 + bp_m)}{V_m} \tag{11}$$

$$Gu_{i,jj} + \frac{2G_v}{1 - 2v}\varepsilon_v - \beta_f p_f - \beta_m p_m - \frac{a\rho_s RT\ln(1 + bp_m)}{V_m} + F_i = 0 \tag{12}$$

where $\sigma_a$ is the adsorption expansion stress, Mpa; $G$ is the shear modulus of coal, MPa; $u_{i,jj}$ is the displacement component, $m$; and $F_i$ is the bulk force, MPa.

## 3. Numerical Model and Parameters for Gas Extraction from Cascade Boreholes

### 3.1. Geometric Models and Boundary Conditions

As shown in Figure 1, the geometric model is derived from a simplification of the site conditions at the working face of the No. 2 coal seam in the Dongpang mine, and a three-dimensional geometric model of a coal seam with a length of 50 m, a width of 40 m, and a height of 5 m is constructed through using COMSOL. The drill hole is located in the middle of the model, with a depth of 50 m and a 114 mm extraction hole diameter. The upper part of the model is set as the stress boundary, whilst the lower part is set as the fixed boundary; the left, right, and rear sides are all rolling boundaries. The diffusive seepage of matrix gas and the seepage of fractured gas affect each other, and they are boundary conditions for each other. When the gas pressure equilibrium within the coal is broken via drilling and extraction, the pressure gradient can be considered equivalent to the volumetric force, and the negative pressure of the borehole can be considered as the boundary controlling the gas flow. The initial gas pressure of the coal seam is 1.5 MPa, and the negative pressure of the extraction hole is 25 kpa, with zero pressure boundary around it. The coal seam model mesh is divided into a quadrilateral mesh with a more refined treatment; the borehole boundary is encrypted with a free triangular mesh to improve the calculation accuracy, and the rest of the locations are swept.

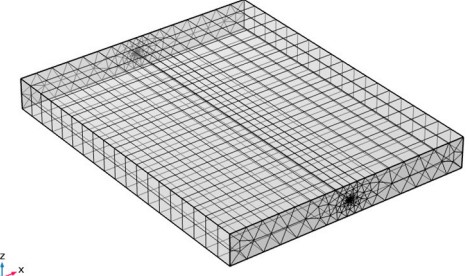

**Figure 1.** Three-dimensional geometry model meshing diagram.

### 3.2. Parameter Assignment

Based on the working face site conditions and test measurements, the physical parameters required for this model are shown in Table 1.

**Table 1.** Simulation parameters.

| Parameter Name | Numerical Value | Parameter Name | Numerical Value |
|---|---|---|---|
| Initial substrate permeability, $k_{m0}$/mD | $1 \times 10^{-4}$ | Langmuir volume product, $a$/(m$^3$/t) | 28.9 |
| Density of the coal body, $\rho_s$ /(kg/m$^3$) | 1500 | Extraction negative pressure, $p_b$/kPa | 25 |
| Initial fracture permeability, $k_{f0}$/mD | 0.1 | Fracture stiffness, $K$/MPa | 4800 |
| Initial gas diffusion coefficient, $D_0$/(m$^2$/s) | $5.6 \times 10^{-12}$ | Poisson's ratio of coal, $\nu$ | 0.33 |
| Dynamic viscosity, $\mu$/ Pa·s | $1.34 \times 10^{-5}$ | Coal skeleton bulk modulus, $K_s$/GPa | 0.166 |
| Limit adsorption deformation, $\varepsilon_L$ | 0.012 | Modulus of elasticity of coal, E/MPa | 2100 |
| Langmuir pressure, $P_L$/MPa | 0.75 | Initial substrate porosity, $\varphi_{m0}$ | 0.06 |
| Attenuation coefficient, $\lambda$/s$^{-1}$ | $4 \times 10^{-17}$ | Initial fracture porosity, $\varphi_{f0}$ | 0.001 |
| Coal seam temperature, T/K | 315.15 | Klinkenberg factor, $K_b$/MPa | 0.76 |

## 4. Analysis of Gas Transport Pattern Simulation Results

### 4.1. Coal Seam Gas Pressure Variations

Considering the dual-medium nature of the coal seam, Figure 2 below shows the variation curve of the fracture and matrix gas pressure in the coal seam for 180 d of extraction. From Section 2.1, it is clear that there is a significant Klinkenberg effect in the process of gas seepage through the coal seam, that coal permeability is directly influenced by gas pressure, and that the Klinkenberg effect helps to facilitate the gas transport process in low-permeability coal seams. For the same extraction time, the fracture gas pressure in the area where the Klinkenberg effect is considered decreases significantly faster than in the area where it is not considered, while the matrix gas pressure changes in the opposite direction to the fracture. This is because, as the extraction time increases, the effective seepage channels within the coal body are reduced, resulting in a blockage of gas diffusion and therefore a slower decline in coal matrix pressure. As the Klinkenberg effect leads to an increase in the molecular thickness of the fissure surface, the source of mass diffusion from the coal matrix into the fissure decreases and the amount of gas within the fissure decreases at a faster rate initially and then gradually slows down as the percolation rate is affected. This pattern therefore reflects the need to consider the Klinkenberg effect.

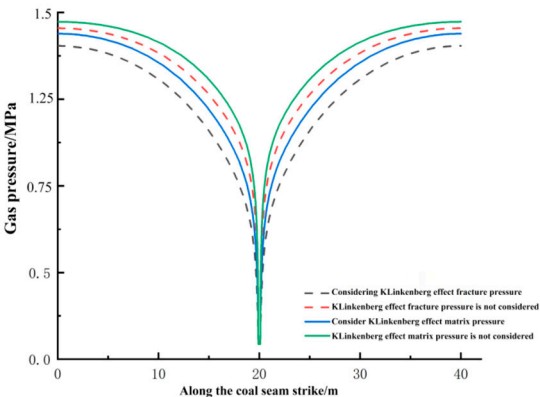

**Figure 2.** Considering the Klinkenberg effect on the coal seam gas pressure.

As can be seen from Figure 3, the pressure difference between the seam pressure and the negative extraction pressure of the borehole at the same extraction time leads to the accumulation of gas farther away from the borehole, such that the further away from the center of the borehole the greater the gas pressure, and at the farther coal wall the initial pressure of the seam is approached. As the extraction time increases, the gas pressure

decreases, with the rate of decline being faster between 30 d and 90 d and decreasing after 90 d of extraction. In the later stages of extraction, the resistance to the diffusion of the matrix gas increases and the source of diffusion mass decreases, which directly leads to a reduction in the amount of gas flowing into the fissure system; the gas pressure in the fissure decreases continuously, and the gas extraction effect tends to stabilize.

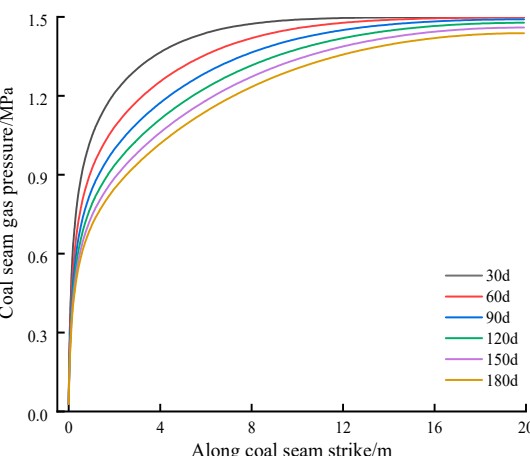

**Figure 3.** Variation in the residual pressure in the coal seam with extraction time.

*4.2. Effective Extraction Radius Simulation*

As seen in relevant studies [25–28], after a coal seam has been pre-sumped by the borehole, the pressure drop area around the borehole when the coal seam gas pressure drops to 75% of the original content within a certain extraction time is the effective extraction influence radius coverage. When it drops to 50% of the original gas pressure, the corresponding extraction reach area is the effective extraction radius coverage.

According to field engineering practice, borehole diameters of 75 mm, 94 mm, 114 mm, and 150 mm were selected in order to compare the effects of different borehole diameters on the radius of gas extraction. The results are shown in Figure 4; it can be seen that the effects of different extraction hole diameters on gas extraction are expressed in the rate of the drop in the gas pressure. In the first 60 d of extraction, the growth rate of the effective extraction radius of the 75 mm borehole was 64%, that of the 94 mm borehole was 67%, that of the 113 mm borehole was 65%, and that of the 150 mm borehole was 66%. With the increase in the extraction time, the gas pressure drop rate was significantly smaller than the initial period after 120 d of extraction, and the pressure drop curve gradually became flatter. The radius of influence of gas extraction from 0 to 120 d was selected for analysis, as can be seen from Figure 5. The radius of influence of extraction expanded at the fastest rate within 60 d of extraction, with the radius of influence of the 75 mm borehole growing at 34%, that of the 94 mm borehole at 60%, that of the 114 mm borehole at 40%, and that of the 150 mm borehole at 40%. The effective radius of influence for the 150 mm borehole was 45%. As the mining depth increases the gas pressure decreases; the effective stress on the coal body increases and the permeability decreases. This being the case, after 60 d the gas pressure decreases at a slower rate, and the growth rate of the effective extraction radius as well as the radius of influence decrease. The effective radius of influence growth rate of a 94 mm borehole is higher than that of a 114 mm borehole under the same conditions, and the use of a 150 mm borehole is known to be too large for the site, increasing the risk of coal to gas. In summary, a borehole diameter of 94 mm was chosen to provide the best extraction results.

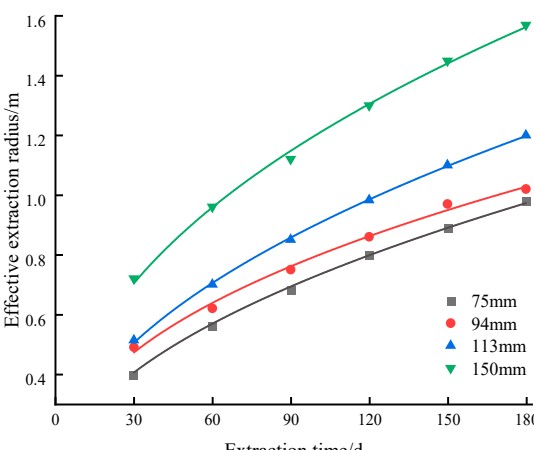

**Figure 4.** Effect of different borehole diameters on the effective extraction radius.

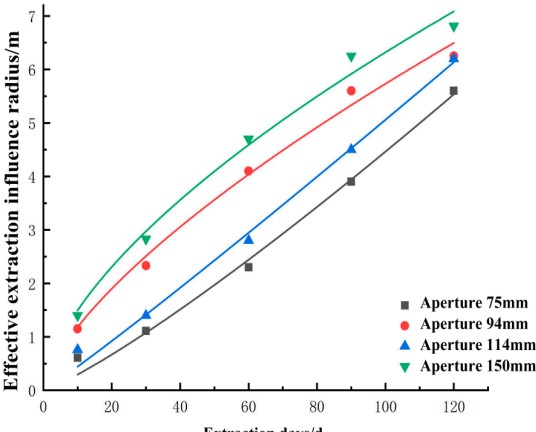

**Figure 5.** Effect of different borehole diameters on the radius of influence of effective extraction.

As can be seen from Figure 6, the extraction radius of the borehole increases with an increase in extraction time: the effective extraction radius and influence radius are 0.3 m and 1.34 m, respectively, at 30 d of extraction, and increase to 1.2 m and 5.9 m, respectively, after 180 d. Fitting the changes within 180 d, it can be seen that the effective extraction radius and extraction time are in accordance with the power exponential function; the specific mathematical relationship is as follows:

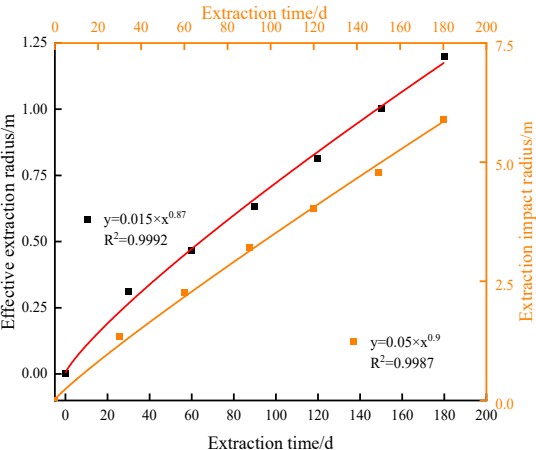

**Figure 6.** Fitted curve of the effective extraction radius of the borehole as a function of time.

$$y_1 = 0.015 \times x^{0.87}, \ R^2 = 0.9992 \tag{13}$$

$$y_2 = 0.05 \times x^{0.9}, \ R^2 = 0.9987 \tag{14}$$

### 4.3. Study of the Main Stress Distribution in Drilled Holes

From the study in [16], it is clear that the principal stresses around the borehole are redistributed as the gas is extracted. Figure 7 shows the distribution of principal stresses around the gas borehole extraction. The drilling of the borehole resulted in the destruction of the coal structure and the formation of an unloading zone around the borehole; therefore, the gas pressure was reduced and the stresses around the borehole were redistributed. The second principal stress distribution shown in Figure 7a is shaped in a similar manner to a bow-tie ripple tail, with a spiral pattern spreading from the center of the borehole up and down the top and bottom plates. The stress at the end of the borehole is the lowest, approximately 0.46 MPa, and the stress at the furthest end of the borehole is the highest, approximately 7.26 MPa. An analysis of Figure 7b shows that the third principal stress distribution around the borehole is shaped in a similar manner to a capsule, spreading outwards from the center of the borehole and forming corrugated spiral knots at both ends of the borehole. The stress around the end of the borehole decreases horizontally and increases vertically, with a minimum value of 1.75 MPa in the vertical direction and a maximum value of 21.8 MPa in the horizontal direction.

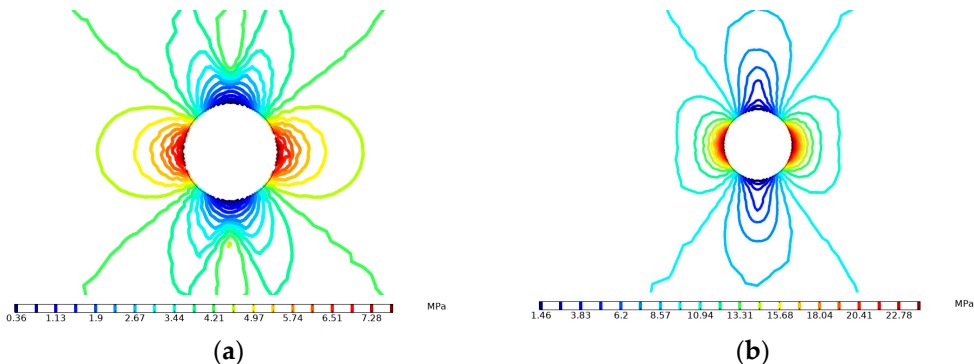

**Figure 7.** Principal stress distribution around the borehole. (**a**) The second principal stress. (**b**) The third principal stress.

### 4.4. Numerical Simulation of the Superimposed Effect of Multiple Boreholes

From the relevant research [10–14], it can be seen that the effective extraction radius, r, area is the main area of the reduction in the coal mine gas pressure, and that the effective influence radius, R, is the secondary area of the reduction in the coal seam gas pressure; the reasonable hole spacing is within the interval (2r, R + r). Based on the results of the previous study on the effective radius of gas, a reasonable hole spacing of (2.4, 7.34) m was obtained, and 3 m, 5 m, and 7 m hole spacings, within the interval, were selected in order to study the gas transport pattern. The geometric model diagram for multi-hole extraction is shown in Figure 8, using parametric scanning to achieve different hole spacing inputs.

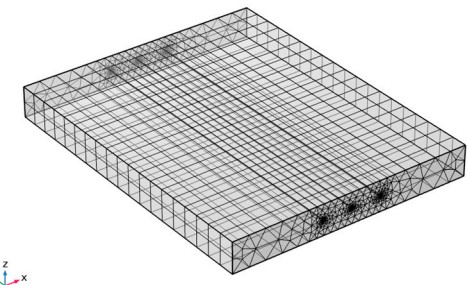

**Figure 8.** Principal stress distribution around the borehole.

A cloud plot of the inner-layer pressure variation at different extraction spacings is shown in Figure 9. After 180 d of extraction, at a spacing of 3 m, the gas pressure between boreholes drops to below 0.46 Mpa; at a spacing of 5 m, the gas pressure between extraction drill holes drops to below 0.52 Mpa; and at a spacing of 7 m, the gas pressure between boreholes drops to below 0.7 Mpa. Therefore, a reasonable range of drill holes is laid out between 2r < D < R + r. For superimposed extraction gaps between adjacent boreholes at the same extraction time, the results of the study support the conclusions of the study in [25]. The smaller the spacing between boreholes, the lower the gas pressure between boreholes and the smaller the residual gas value of the coal seam.

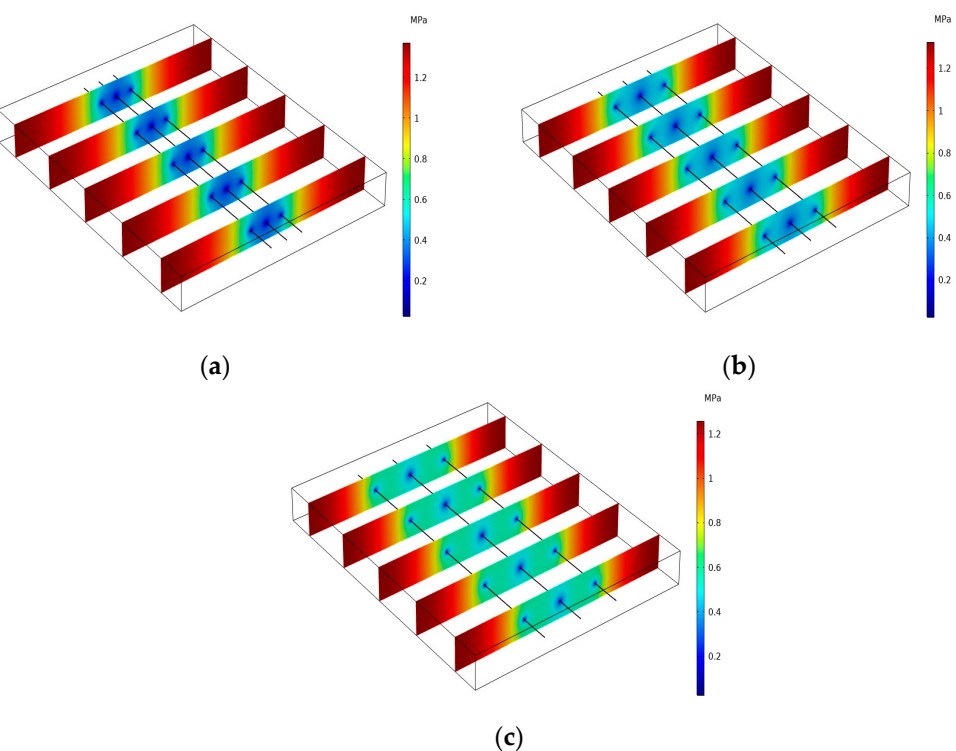

**Figure 9.** Cloud plot of gas pressure variation at different spacings for 180 d of extraction. (**a**) Drill hole spacing of 3 m. (**b**) Drill hole spacing of 5 m. (**c**) Drill hole spacing of 3 m.

### 4.5. Coal Body Permeability Variation Pattern

Reference studies [7–9] show that the coal seam diffusion coefficient varies dynamically with the attenuation coefficient at different extraction times, and that the attenuation coefficient actually reflects the transfer process from the outer pores of the coal to the inner fractures, with larger values reflecting a more difficult transition between pore fractures. As can be seen from Figure 10, the diffusion coefficient of the coal seam changes dynamically with the attenuation coefficient at different extraction times. The attenuation coefficient actually reflects the transfer process from the outer pores of the coal to the inner fractures,

with larger values reflecting a more difficult transition between pores and fractures. As extraction proceeds, the ratio of gas diffusion coefficients decreases, with the diffusion coefficient decreasing rapidly in the early stages of extraction and tending toward zero in the later stages until the end of extraction. This phenomenon confirms the need to consider dynamic diffusion coefficients and provides a theoretical basis for subsequent research into the variation in coal seam permeability.

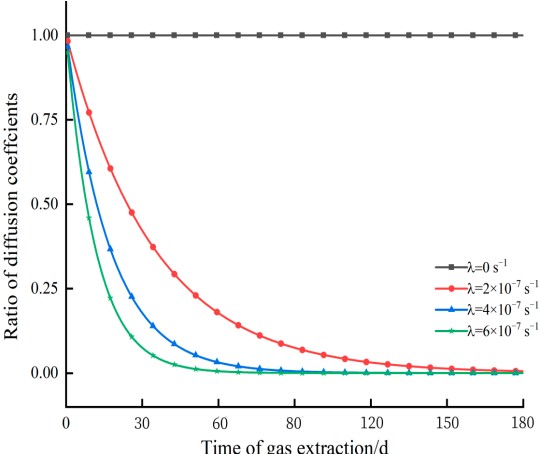

**Figure 10.** Variation curve of the diffusion coefficient with the extraction time for different attenuation coefficients.

Figure 11 shows the effect of the matrix gas diffusion coefficient on coal permeability at different extraction times, with $\lambda = 0$ acting as an indication that the coal seam gas diffusion coefficient is a constant, while three different sets of attenuation coefficients are taken for the simulation. The effect of the attenuation coefficient on the coal seam permeability was not significant at a pumping time of 10 d. As the extraction time increases, the change becomes obvious at 60 d–90 d, and the decreasing trend in permeability increases with an increase in the attenuation coefficient, and the change trend becomes stable after 120 d of extraction. The reason for this phenomenon is that, as the extraction time increases, the diffusion coefficient decreases as the decay coefficient increases, hindering the diffusion of gas from the pore space to the fissure. The increase in resistance to gas diffusion leads to a decrease in the amount of gas flowing into the fissure system, a decrease in gas pressure within the fissure, a decrease in the effect of matrix shrinkage on permeability, and a decrease in coal permeability due to the increase in effective stress on the coal body.

### 4.6. Simulation of Seepage Velocity Patterns

As shown in Figure 12, the change in the gas seepage velocity with time can be roughly divided into three stages: rapid rise, slow fall, and stable and constant. The results of this study are consistent with the pattern of gas seepage velocities in fractures in the literature [11–13]. As can be seen from the research in Section 4.1, the process of gas transport in the coal seam is directly influenced by the matrix mass source, and the gas seepage is closely related to the pressure difference between the matrix and the fissure, with the pressure difference changing as shown in Figure 13. At the early stage of extraction, due to the large pressure difference between the fissure pressure and the negative extraction pressure, the free gas in the fissure system rapidly enters the extraction borehole, resulting in the gas seepage rate in the fissure rapidly rising to the peak. As the extraction time increases, the Klinkenberg effect leads to an increase in the molecular thickness of the fissure surface and a reduction in the mass source from the coal matrix diffusing into the fissure; the pressure differential gradually decreases to a stable value during the 120 d–180 d extraction cycle and the gas seepage rate in the fissure begins to slowly decrease. When the mass source provided by the matrix is not sufficient to replenish the gas flowing out of the fissure,

the pressure difference between the fissure and the outside of the borehole decreases, and the seepage rate will continue to slow to a plateau value.

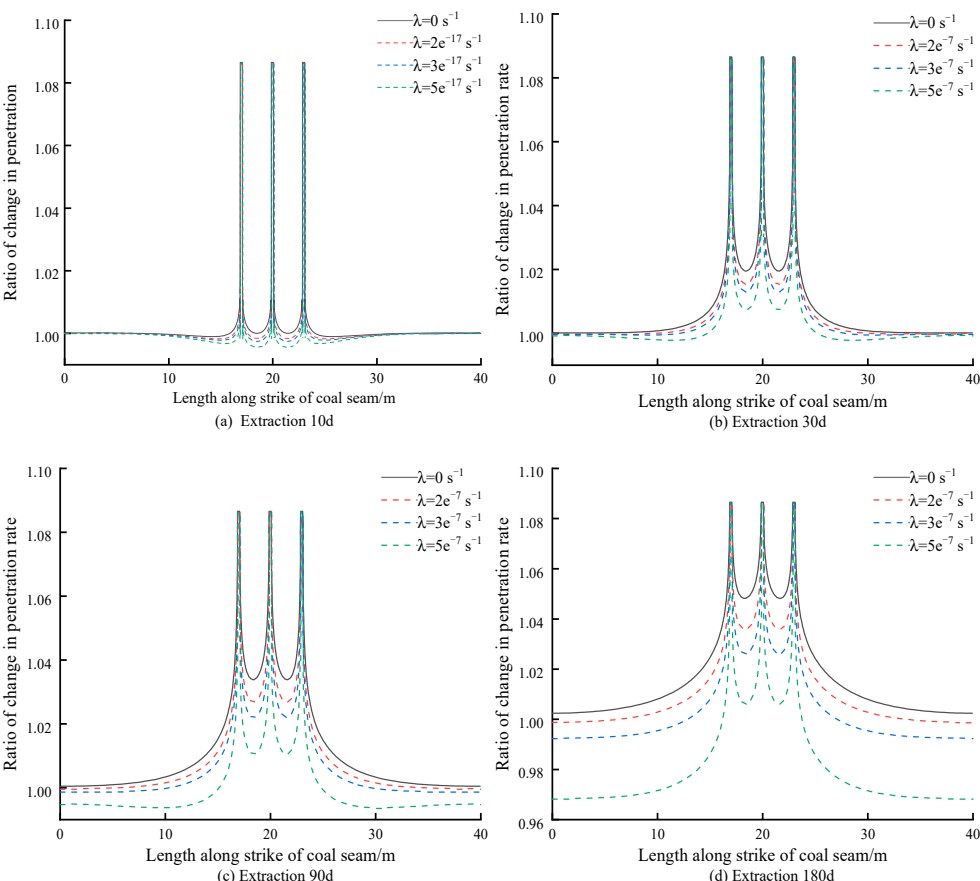

**Figure 11.** Decay of coal permeability with increasing extraction time.

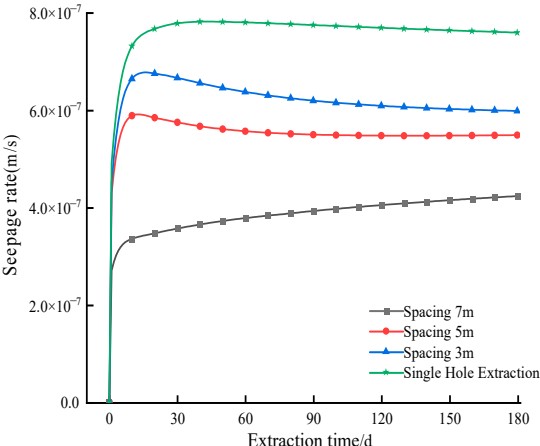

**Figure 12.** Variation in the gas seepage velocity at different borehole spacings.

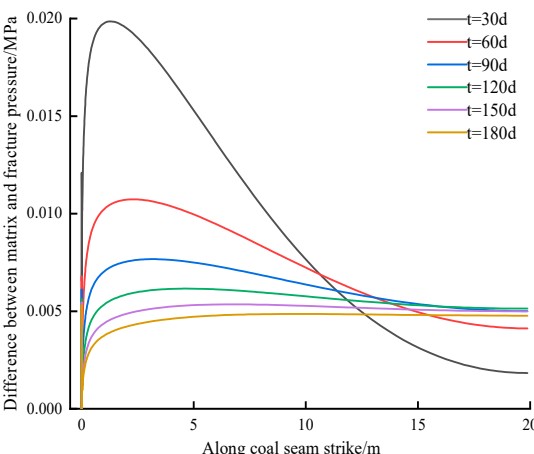

**Figure 13.** Curve of pressure difference between matrix fractures as a function of time.

## 5. Optimization of Hole Layout Considering Superposition Effects

### 5.1. Extraction Compliance Time

The effective extraction radius of 1.2 m after 180 d of single-hole extraction was taken as the condition, and the coordinate point (1.2, 0) was arranged as the monitoring point with which to study the relationship between residual gas pressure and extraction time under different drill hole spacing; the results are shown in Figure 14. When the spacing between boreholes was 5 m and 7 m, the difference in the drop in pressure was not obvious as the interaction between boreholes was small at the beginning of extraction, resulting in a small difference with single-hole extraction; after 30 d of extraction, the superimposed cross-extraction interaction between boreholes became more and more significant with the extraction time. The gas pressure drop rate increased continuously within 30–120 d and gradually stabilized at the end of extraction, which correspond to the previous study on gas seepage rates. For the 3 m spacing, the effect of inter-borehole extraction is more pronounced in the early stages of extraction. When the spacing between boreholes is too small, there may be overlapping areas of extraction and the amount of borehole construction may increase, raising the costs of gas extraction. From Figure 15, the extraction time was determined by using the prescribed pressure of 0.74 MPa as a boundary with which to determine the extraction compliance value according to the measurement criteria found in the literature [13–15]. The extraction compliance times were 25 d, 35 d, and 46 d for borehole spacings of 2 m, 3 m, and 4 m, respectively, and 57 d, 66 d, and 75 d for borehole spacings of 5 m, 6 m, and 7 m, respectively, which are consistent with the effective extraction radii over time for single-hole extraction conditions.

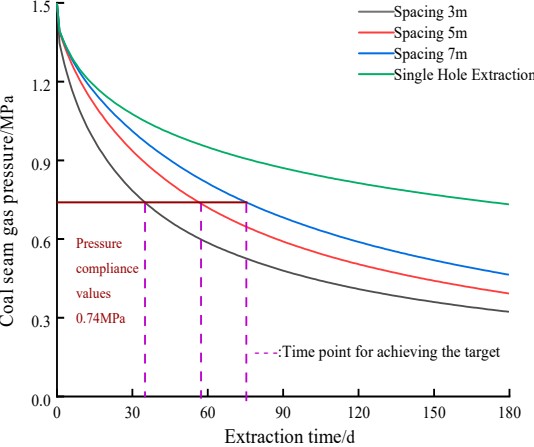

**Figure 14.** Variation in the gas pressure in the center of the borehole at different spacings.

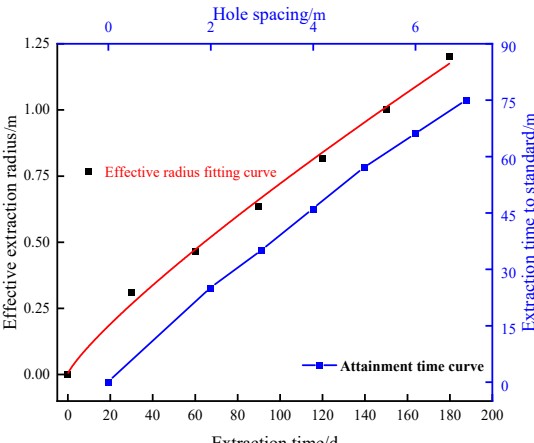

**Figure 15.** Gas extraction compliance curve vs. effective radius at different spacings.

### 5.2. Optimization of Multi-Row Down-Plunge Drilling

From the study in [9], it can be seen that the "positive triangle" method of hole placement ensures that the pressure at the coal wall from the center of the borehole is within the specified range, while avoiding the influence of superimposed extraction blind zones between multiple holes. As the effective extraction area of the coal seam is a circular area with the borehole as the center and the effective extraction radius, r, as the radius, blindly increasing the number of boreholes will not only result in a waste of manpower and material resources, but also increase the difficulty of the borehole construction process. Additionally, it will not avoid the formation of extraction blind zones between boreholes, which will also increase the possibility of coal and gas protrusion. In view of this, the blind zone cannot be eliminated by taking two times the effective radius as the spacing between boreholes, and too many boreholes will increase the risk; this being the case, a positive triangular arrangement of boreholes should be used, as shown in Figure 16.

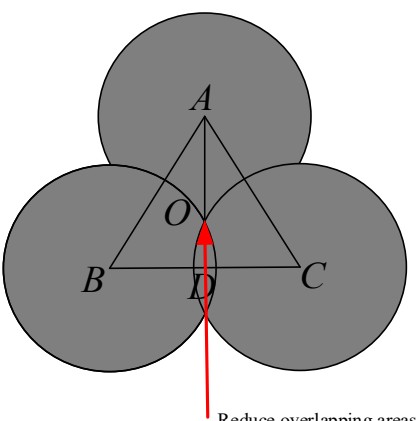

Reduce overlapping areas

A:Drilling；O:Triangular center；B:Drilling；C:Drilling；D:Stacking region

**Figure 16.** Diagram of the theoretically optimized hole layout scheme.

The boreholes are arranged in parallel and perpendicular stratigraphic directions, such that they form equilateral triangles with each other in order to eliminate the extraction blind zone between the boreholes. The height, H, of the triangle is the distance between the holes, and the spacing, L, is the length of the triangle, which can be calculated from the equation as 1.8 m and 2.1 m for H and L, respectively. The construction of the workings at the site is carried out according to this plan, and the schematic diagram of the plan is

shown in Figure 17, which ensures the effective extraction range while reasonably reducing the negative impact of the stacked area. The calculation is shown below:

$$H = 1.5r$$

$$L = \frac{H}{sin\frac{\pi}{3}}$$

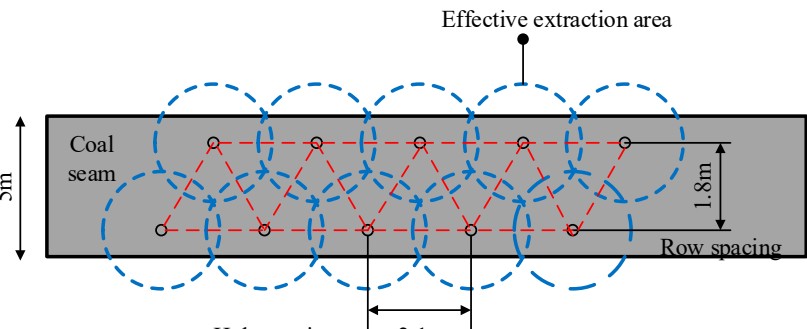

**Figure 17.** Illustration of the positive triangular hole layout in the extraction face.

### 5.3. Numerical Simulation of Square Triangular Layout Holes

On the basis of the above coupled model, the 2D working face model section was intercepted and the reference center line, AB, was arranged in the geometric model, as shown in Figure 18. Combined with the feedback from the field engineering test, the extraction effect of the positive triangular arrangement method was verified by arranging the upper and lower rows of holes, as shown in Figure 19. As can be seen from Figure 19, after optimizing the hole layout, the gas pressure decreases in a concave pattern with an increase in the extraction time along both sides of the borehole, and the minimum value of the gas pressure occurs at the center of the borehole; the trend in the coal seam gas pressure variation curve is consistent with the simulation results from the study in [25]. This method effectively solves the disadvantage of high gas pressure between boreholes in the common method of hole placement.

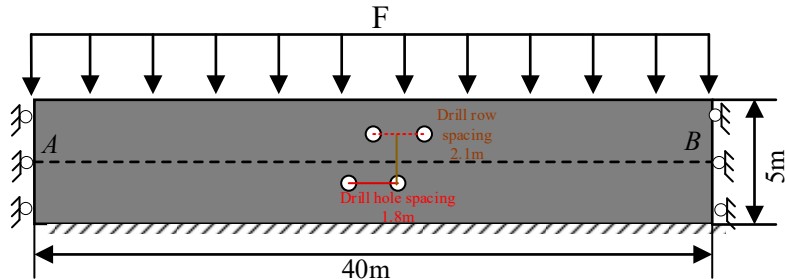

F:Overburden stress of coal seam; AB:center line

**Figure 18.** Simulated geometric model after hole layout optimization.

### 5.4. Field Engineering Application

The technical parameters for gas extraction applied on-site in this project were based on the original technical parameters for the working face, with changes to the borehole diameter, borehole spacing, and hole layout, while other extraction parameters remained unchanged. Based on the mining succession arrangement and the simulation results of this paper, and with reference to the field operability of the study [26–28], the final arrangement of gas pre-sumping boreholes at the coal mining face was determined. Five observation holes were set at distances of 1 m at the center between any boreholes, along the center line direction. After optimization, the pure gas extraction volume reached 0.614 m³/min;

compared with the previous average pure extraction volume of 0.405 m³/min, the pure extraction volume increased by 51.5%. As shown in Figure 20, the drop in pressure of the five observation holes all exceeded 12%, and all five observation holes were within the effective influence extraction radius, which is consistent with the simulation results of this paper, achieving a good extraction effect and realizing the safe prevention and reasonable use of gas resources.

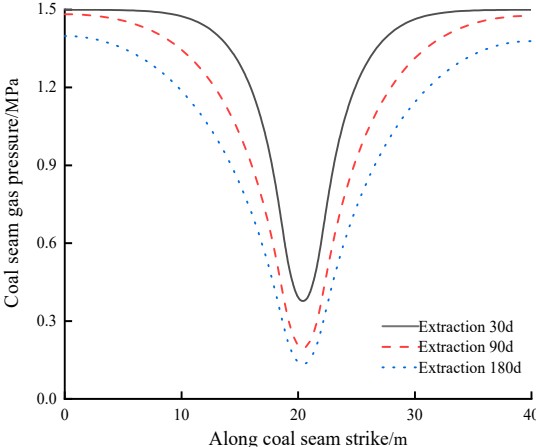

**Figure 19.** Plot of gas pressure over time in a positive triangular borehole.

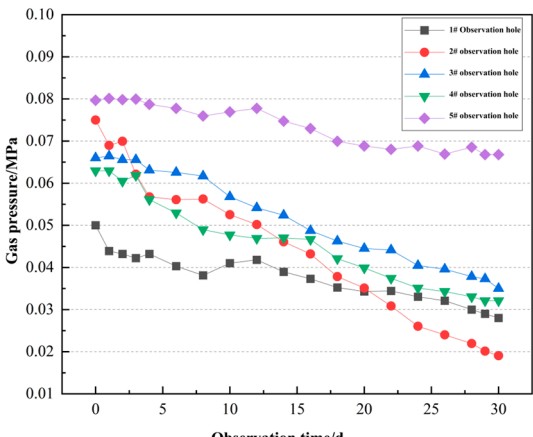

**Figure 20.** Graph of the gas pressure in the observation holes at the working face over time.

## 6. Discussion

In this paper, a coal body is regarded as a homogeneous elastic medium with double pores, fissures, and double permeability. On the basis of considering gas adsorption and desorption, the Klinkenberg effect, the matrix shrinkage effect, and permeability evolution, as well as introducing the dynamic gas diffusion coefficient, the coal matrix porosity model and fissure porosity model, in addition to the the coal matrix permeability model and fissure permeability evolution model, are constructed, respectively, combined with the coal bed deformation equation and the flow–solid coupling model of double-hole double-percolation flow in gas-bearing coal seams. The established multi-field coupling relationship is shown in Figure 21.

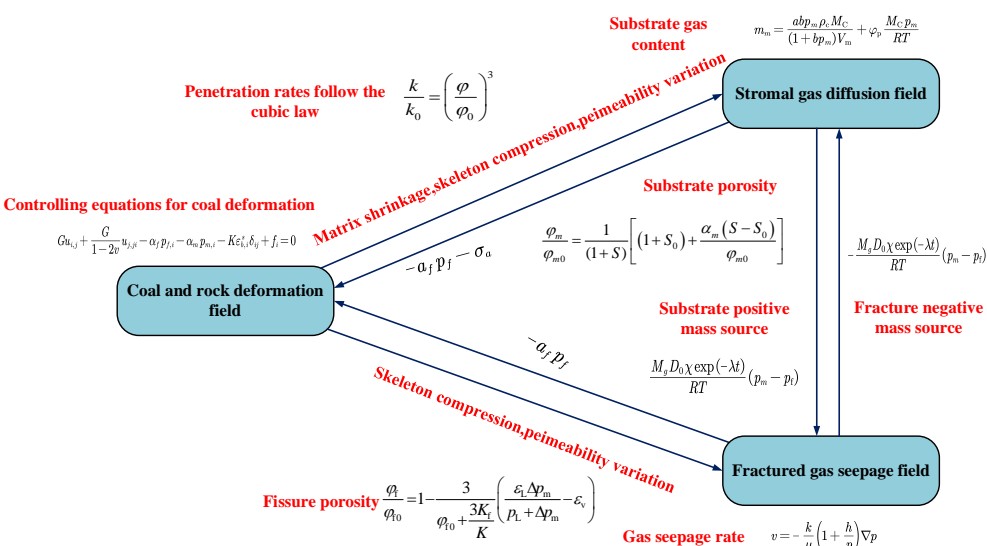

**Figure 21.** Multi-field coupled relationship diagram considering the dual permeability of coal.

Most previous studies have treated the structure of coal as a dual-pore–single-permeability model, ignoring the seepage movement within the coal matrix system. Therefore, this paper adopts the theory of the dualpore–dual-permeability model, theorizing the structure of coal as a dual-media model of pore–fissure and considering the influence of matrix permeability as well as fissure permeability on gas transport. Gas transport in a coal seam can be considered to be a two-step process in which diffusion and seepage occur simultaneously, i.e., gas in the coal matrix not only enters the fissures via diffusion, but also moves between the matrix in the form of seepage and is eventually discharged through the roadway or borehole, also considering the reverse effect of the gas in the fissures on the diffusion and mass transfer rates of the coal matrix when influenced by the seepage movement of the matrix gas. During this process, matrix diffusion continuously shows decaying changes with time, and in turn acts as a mechanism for changes in coal permeability as well as gas pressure. The change in permeability is a result of the competing effects of the matrix shrinkage and skeletal deformation of the coal body. As the extraction time increases, the matrix gas pressure around the borehole gradually decreases and the matrix shrinkage effect gradually dominates, leading to an increase in the coal seam permeability; the gas seepage rate at the observation point can be divided into three stages: a fast rising stage, a slow falling stage, and a stable and constant stage.

The model is used to study and analyze the changes in the seam pressure, effective extraction radius, seepage velocity, and coal permeability during the pre-pumping process, which helps to accurately grasp the coal seam gas transport pattern. Finally, the study was combined with field engineering practice to optimize the hole placement method at the Dongpang mine, proposing a suitable hole size and a square triangular hole placement method, which successfully eliminated the extraction blind zone caused by the superposition effect of multiple holes. The results of this study can be applied to the deep mining of coal seams with low permeability, and the use of the square triangle method as a hole placement method for gas extraction work can effectively eliminate the overlapping phenomenon in the area affected by gas extraction and eliminate the extraction blind zone, which can reduce or avoid the possibility of coal and gas protrusion as well as ensure the safe mining of coal resources. Future research will focus on the effects of plastic damage and temperature on the coal body for more accurate guidance on gas extraction, in addition to the promotion of efficient and safe production in the coal mining industry.

## 7. Conclusions

1. On the basis of the matrix seepage field of double seepage—diffusion, a coal seam gas–solid coupling model was established in combination with the coal deformation

field. Through COMSOL simulation, it can be determined that the gas pressure drop rate decreases continuously with an increase in the extraction time to stabilization, and that the main stress around the borehole redistributes. The variations in seepage velocity show different patterns with time of extraction, and a larger diffusion attenuation coefficient leads to the weakening of the matrix contraction effect as well as an accelerated rate of permeability decline.

2. A reasonable hole diameter for drilling is 94 mm, the effective extraction radius, r, is 1.2 m, and the influence radius, R, is 5.9 m. The spacing of holes was studied in the range of 2r < d < R + r. There is a blind extraction zone between adjacent holes due to the superimposed effect of extraction; the pressure between holes decreases with a decrease in the spacing, and the superimposed cross-extraction effect gradually becomes obvious with an increase in time.

3. Considering safety and economic factors, and reducing the negative impact of blindly reducing the drill hole spacing, the equilateral triangular extraction method was used for the Dongpang mine. The drill hole spacing was 1.8 m and the drill hole row spacing was 2.1 m, effectively avoiding pressure extraction overlap zones and blind areas between drill holes. After pre-pumping verification on-site, the pressure drop in the observation holes all reached the prescribed level, reducing the risk of coal and gas protrusion and providing a theoretical basis for gas extraction engineering designs.

**Author Contributions:** Writing—original draft preparation and collecting field data, J.Y.; provision of experimental protocols and geometric modeling, M.Z.; major translation work and model commissioning, W.Z.; data processing and drawing, Q.K. All authors have read and agreed to the published version of the manuscript.

**Funding:** This research was funded by the National Natural Science Foundation of China (52174086), the National Natural Science Foundation of China Youth Fund (51804222), the 14th Graduate Innovative Fund of the Wuhan Institute of Technology (CX2022577), the 2022 Hubei Master Teacher Studio, and the Key Project of Hubei Province Education Department (D20201506).

**Institutional Review Board Statement:** Not applicable.

**Informed Consent Statement:** Not applicable.

**Data Availability Statement:** Not applicable.

**Conflicts of Interest:** The authors declare no conflict of interest.

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
