# Peer review of "Simulation Study on the Characteristics of Gas Extraction from Coal Seams Based on the Superposition Effect and Hole Placement Method"

_sustainability, doi:10.3390/su15108409_

Round 1

Reviewer 1 Report

The article combines field and numerical simulations to investigate the gas transport pattern of coal seams and proposes a new extraction method with higher efficiency. The significance of the study is relatively new and is recommended for acceptance after minor revision.

1. Check the explanation of variable characters throughout the text, some variables in Equation 2 are incorrectly labelled and the explanation of Equation 7 is incorrectly formatted and should be carefully revised.

2. The meaning of the Klinkenberg effect should be explained in the article when studying the variation in gas pressure in section 4.1.

3. The lines in Fig. 7 are in as dark a colour as possible to allow for clear publication printing.

4. The legends in the figures accompanying the article should be supplemented with meanings, such as l in Figures 9 and 10.

5. In section 4.2: "After the coal seam has been pre-smoked by the borehole for a period of time i.e. the effective mining radius is covered". The statement is cumbersome to express and needs to be modified.

No

Author Response

Thank you for handling my manuscript during your busy schedule. I have followed the review comments line by line and the revision notes and red marked revisions have been uploaded in the form of attachments.

Reviewer 2 Report

This paper presents simulation study on the characteristics of gas extraction from coal seams based on the superposition effect and hole placement method.  

line 51 : Wang et al[10] should be write Wang et al. [10]

line 54 : Chen [11] et al. introduced should be write Chen et al. [11]

line 60, 63, 66, 69, 72, 75,77, 80, 84,89 : please check citation formatting style

line 164-175 : please check formatting (eg: justifiy not align left style)

line 184 : the Figure. 2 below should be write "the Figure 2 below"

line 225 : and the pressure The pressure drop curve ...please check typo

line 339 : in the Figure.12. When should be write "in the Figure 12. When"

General comments:

1. Abstract : Should mention the methodology used in the study

2. Some formatting errors were spotted (please refer to the suggestions for authors section above)

3. It might be good if the authors be able to support their finding and discussion with previous work by other researchers since this work was solely using the numerical modeling especially in Section 4.3, 4.4, 4.5, 4.6, 5.1, 5.3 and 5.4.

4. Discussion section is suggested to be improve and elaborate. 

5. All references were up to date (< 5 years)

Minor editing is needed.

Author Response

Thank you for handling my manuscript during your busy schedule.  I have revised the manuscript line by line in accordance with the review comments and the revision notes and red marked revisions have been uploaded via a separate attachment.

Reviewer 3 Report

This article uses COMSOL to simulate the mechanism of the superposition effect on the transport of gas in coal seams and proposes a new way of hole placement. The article is of good quality and suitable for publication, and the following are some revisions.

1.What is the basis to establish the physical model in the numerical simulation? And how to control the boundary conditions of the model? It would be better to add related contents.

2. Some of the parameters of the equations in the paper lack explanation, it is recommended that the meaning of the relevant parameters in equations 2, 5, 6, 7 and 8 be explained in detail.

3. The units of µ are missing from the simulation parameters in section 3.2, table 1, and it should be supplemented and the other variables in the table double-checked.

4. Check the full text that the graph name is accurate and corresponds to the narrative, e.g. "Figure 10: Variation of coal permeability decay at different extraction times " should be changed to "Decay of coal permeability with increasing extraction time".

5. The practical value of the research can be added to the discussion to experience the operational aspects, such as how the research results can be applied to the coal mining industry.

6. It is recommended that the value of the research be sublimated in conclusion 3, which should focus on the importance of the findings.

The quality of English should be minor revised.

Author Response

Thank you for taking time out of your busy schedule to deal with my manuscript. I have revised the manuscript line by line in accordance with the review comments and the revision notes and red marked revisions have been uploaded via a separate attachment.

Reviewer 4 Report

This paper establishes a fluid-solid coupling model for dual penetration of pore fractures in coal seams and determines the best hole placement method that can eliminate the extraction blind zone. It has been applied well in the field and is suitable for publication. I suggest a minor revision of this paper. Some comments for revision are given below.

(1) The literature review on gas-solid coupling models in the Introduction suggests adding relevant literature on diffusion permeability models for gas-bearing coal.

(2) What aspects of the numerical simulation in this thesis were considered in building the physical model? And what are the advantages of using this model to achieve accurate gas extraction placement techniques?

(3) The physical quantities represented by the variable characters in the full text should be given an exact explanation when they first appear. For example, what is the meaning of equation (1) eL?

(4) The study of gas seepage velocity patterns should be echoed in section 4.1 when modelling the mechanism of the Klinkenberg effect on pressure differentials, and a link can be made between the two. It is suggested that the study of the effect of pressure difference on gas seepage velocity should be added to section 4.5.

(5) When studying the superimposed effect of multiple boreholes in section 4.4, a numerical model model plot of multiple boreholes should be added to be more intuitive.

(6) The format of the reference section should be in a uniform standard format; some of the literature is inconsistently formatted and it is recommended to check and revise it carefully.

Author Response

(The authors gave the same response as above.)

Round 2

Reviewer 1 Report

No

Reviewer 2 Report

Kindly check the formatting error in para 2, page 18.

Reviewer 3 Report

Accept in present form.

Reviewer 4 Report

I am very satisfied with the paper revision.